# AutoLoRa: An Automated Robust Fine-Tuning Framework

**Xilie Xu**[1] **Jingfeng Zhang**[2,3,†] **Mohan Kankanhalli**[1]

[1] School of Computing, National University of Singapore
[2] School of Computer Science, The University of Auckland
[3] RIKEN Center for Advanced Intelligence Project (AIP)
`{xuxilie, mohan}@comp.nus.edu.sg, jingfeng.zhang@auckland.ac.nz`

## Abstract

Robust Fine-Tuning (RFT) is a low-cost strategy to obtain adversarial robustness in downstream applications, without requiring a lot of computational resources and collecting significant amounts of data. This paper uncovers an issue with the existing RFT, where optimizing both adversarial and natural objectives through the feature extractor (FE) yields significantly divergent gradient directions. This divergence introduces instability in the optimization process, thereby hindering the attainment of adversarial robustness and rendering RFT highly sensitive to hyperparameters. To mitigate this issue, we propose a low-rank (LoRa) branch that disentangles RFT into two distinct components: optimizing natural objectives via the LoRa branch and adversarial objectives via the FE. Besides, we introduce heuristic strategies for automating the scheduling of the learning rate and the scalars of loss terms. Extensive empirical evaluations demonstrate that our proposed automated RFT disentangled via the LoRa branch (AutoLoRa) achieves new state-of-the-art results across a range of downstream tasks. AutoLoRa holds significant practical utility, as it automatically converts a pre-trained FE into an adversarially robust model for downstream tasks without the need for searching hyperparameters. Our source code is available at the GitHub.

## 1 Introduction

With the emergence of foundation models (Bommasani et al., 2021), fine-tuning the pre-trained feature extractor (FE) has become a low-cost strategy to obtain superior performance in downstream tasks. Notably, GPT-3 (Brown et al., 2020) can achieve state-of-the-art (SOTA) performance on GLUE benchmarks (Wang et al., 2018) via parameter-efficient fine-tuning (Hu et al., 2021). Due to the ubiquitous existence of adversarial attacks (Goodfellow et al., 2014; Madry et al., 2018; Xu et al., 2022), adopting pre-trained FEs to safety-critical downstream areas such as medicine (Buch et al., 2018) and autonomous cars (Kurakin et al., 2018) necessitates the strategy of robust fine-tuning (Hendrycks et al., 2019) that can yield adversarial robustness in downstream applications.

Robust fine-tuning (RFT) (Hendrycks et al., 2019) that contains an adversarial objective to learn features of adversarial data (Madry et al., 2018) can gain adversarial robustness in downstream tasks. To further improve generalization, vanilla RFT (formulated in Eq. 1, shown in the left panel of Figure 1c) optimizes both adversarial and natural objectives to learn the features of adversarial and natural data simultaneously via the FE (Zhang et al., 2019; Shafahi et al., 2019; Jiang et al., 2020). Recently, TWINS (Liu et al., 2023) (formulated in Eq. 2) further enhances performance in downstream tasks by incorporating vanilla RFT with a dual batch normalization (BN) (Xie et al., 2020; Wang et al., 2020) module. TWINS takes advantage of extra information from the pre-trained FE (i.e., pre-trained statistics in a frozen BN) via the dual BN module, thus improving performance.

However, we empirically find that vanilla RFT and TWINS have a common issue, where optimizing both adversarial and natural objectives via the FE leads to significantly divergent gradient directions. As shown in Figure 1a, the cosine similarity between the gradient of natural and adversarial objective

---

† Corresponding author.

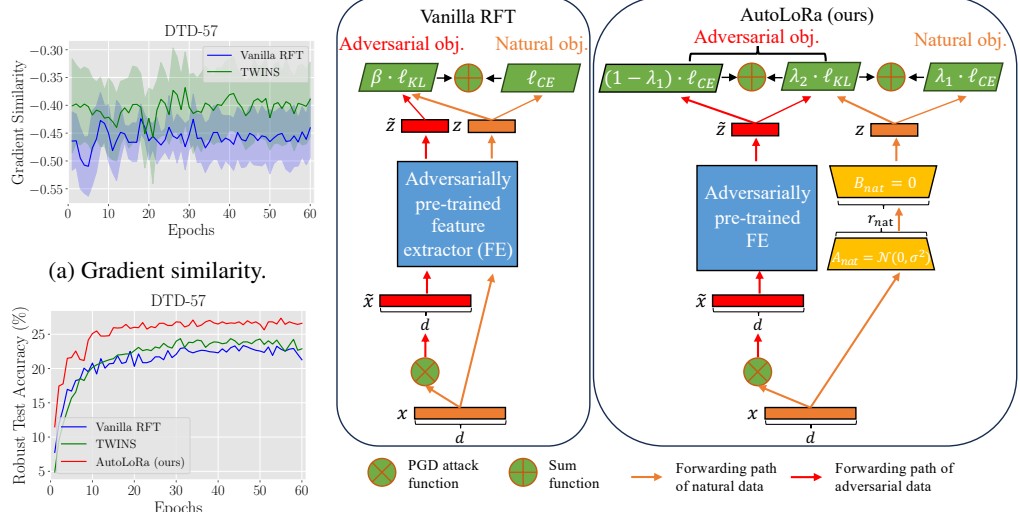

(a) Gradient similarity.

(b) Adversarial robustness.

(c) Left panel: Vanilla RFT. Right panel: Our proposed automated RFT disentangled via a low-rank branch (AutoLoRa). The yellow module is the LoRa branch. Scalars $\beta, \lambda_1, \lambda_2$ serve as reweighting loss terms.

Figure 1: Figure 1a shows the cosine similarity between the gradients of natural and adversarial objective w.r.t. the feature extractor (FE) (dubbed as gradient similarity) on DTD-57 (Cimpoi et al., 2014). Figure 1b shows the robust test accuracy evaluated via PGD-10 on DTD-57. Extra empirical results are shown in Figures 2a and 2b (Appendix B.1). Figure 1c shows the framework of vanilla RFT that learns both adversarial and natural data via the FE while our proposed AutoLoRa learns adversarial and natural data via the FE and the LoRa branch, respectively.

w.r.t. the FE, dubbed as gradient similarity, achieved by vanilla RFT (blue lines) and TWINS (green lines) is very low. It indicates that optimizing both natural and adversarial objectives through the FE can result in a divergent and even conflicting optimization direction.

The divergent optimization directions make the optimization process of RFT unstable, thus impeding obtaining robustness in downstream tasks and making RFT sensitive to hyperparameters. Compared to TWINS, vanilla RFT has a lower gradient similarity (in Figure 1a) while gaining a lower robust test accuracy (in Figure 1b). It indicates that the issue of divergent optimization direction could prevent gaining adversarial robustness in downstream tasks. TWINS tackles this issue to some extent via the dual BN module while gaining only slightly improved robustness. Thus, we conjecture that mitigating the aforementioned issue can further enhance adversarial robustness.

To this end, we propose to disentangle RFT via a low-rank (LoRa) branch (Hu et al., 2021) (details in Section 4.1). As shown in the right panel of Figure 1c, we disentangle the RFT by forwarding adversarial and natural data through the FE (blue module) and the LoRa branch (yellow module), respectively. In this way, the FE parameters are updated only by the adversarial objective to learn the features of adversarial data, which exactly solves the aforementioned issue of the divergent optimization direction. Besides, the FE also learns the knowledge of the natural objective via minimizing the Kullback-Leibler (KL) loss between adversarial logits and natural soft labels provided by the LoRa branch to avoid degrading generalization. Therefore, benefiting from the parameter-efficient LoRa branch, RFT disentangled via a LoRa branch can be a low-cost strategy to further improve adversarial robustness while maintaining generalization.

Moreover, we propose heuristic strategies of automatically scheduling the learning rate (LR) as well as the scalars $\lambda_1$ and $\lambda_2$ (details in Section 4.2). Lin et al. (2019) analogized the generation process of adversarial data to the training process of the neural model. It motivates us to employ the automatic step size scheduler in AutoAttack (Croce & Hein, 2020), which is proven to enhance the convergence of adversarial attacks, for scheduling the LR, thus helping the convergence of RFT. Inspired by graduated optimization (Hazan et al., 2016; Hulse et al., 2019), we take $\lambda_1$ and $\lambda_2$ to be negatively and positively proportional to the standard accuracy of natural training data, respectively. In this way, our proposed scheduler will enforce the model to first focus on solving the natural objective which is a simplified optimization problem and then tackling the difficult optimization

problem of the adversarial objective. Therefore, our automated scheduler of the scalars utilizes graduated optimization to improve robustness.

Our comprehensive experimental results validate that our proposed automated robust fine-tuning disentangled via a LoRa branch (AutoLoRa) is effective in improving adversarial robustness among various downstream tasks. We conducted experiments using different robust pre-trained models (ResNet-18 and ResNet-50 adversarially pre-trained on ImageNet-1K (Salman et al., 2020)) and both low-resolution (Krizhevsky, 2009) and high-resolution (Cimpoi et al., 2014; Khosla et al., 2011; Wah et al., 2011; Griffin et al., 2007) downstream tasks. Empirical results validate that our proposed AutoLoRa can consistently yield new state-of-the-art adversarial robustness performance without tunning hyperparameters compared to TWINS (Liu et al., 2023).

## 2 RELATED WORK

Here, we introduce the related work of fine-tuning and robust fine-tuning.

**Fine-tuning.** With recent advances in self-supervised pre-training (Chen et al., 2020a;b), fine-tuning foundation models via self-supervision on large-scale unlabelled datasets can efficiently achieve powerful performance in downstream tasks. As the number of parameters of pre-trained FE grows significantly, it requires the parameter-efficient fine-tuning (PEFT) strategy that decreases the trainable parameters during fine-tuning. One popular strategy is to introduce an adapter layer during fine-tuning (Houlsby et al., 2019; Lin et al., 2020); however, it brings extra inference latency. Recent studies propose to freeze the pre-trained FE parameters and inject trainable decomposition matrices that are low-rank and thus parameter-efficient (Hu et al., 2021; Chavan et al., 2023). Notably, only fine-tuning low-rank matrices for GPT-3 (Brown et al., 2020) can achieve the SOTA performance on GLUE benchmarks (Wang et al., 2018) without incurring inference latency (Hu et al., 2021; Chavan et al., 2023).

**Robust fine-tuning (RFT).** RFT (Shafahi et al., 2019; Hendrycks et al., 2019) is a low-cost strategy to obtain adversarially robust models in downstream tasks by fine-tuning the pre-trained FEs on adversarial training data (Madry et al., 2018; Zhang et al., 2020; 2021; Chen et al., 2022). To further improve generalization in downstream tasks, recent studies propose to learn the features of natural and adversarial data together (i.e., vanilla RFT) (Zhang et al., 2019; Shafahi et al., 2019). Note that vanilla RFT has been widely applied to fine-tuning adversarially self-supervised pre-trained models (Jiang et al., 2020; Fan et al., 2021; Zhang et al., 2022; Yu et al., 2022; Xu et al., 2023b;c) and achieved powerful robustness in downstream tasks. Furthermore, various strategies about how to utilize extra information from the pre-trained FEs for improving performance have been proposed. Liu et al. (2022) proposed to jointly fine-tune on extra training data selected from the pre-training datasets and the whole downstream datasets to further improve RFT. TWINS (Liu et al., 2023) is the existing SOTA RFT method that incorporates vanilla RFT with a dual BN framework (Xie et al., 2020). TWINS uses the dual BN framework to take advantage of the pre-trained statistics in a frozen BN branch, which is the extra useful information of the pre-trained FEs, thus resulting in superior performance.

## 3 A CLOSER LOOK AT VANILLA RFT AND TWINS

In this section, we first introduce preliminaries of vanilla RFT (Zhang et al., 2019; Jiang et al., 2020) and TWINS (Liu et al., 2023). Then, we empirically disclose the issues of vanilla RFT and TWINS.

### 3.1 PRELIMINARIES

Let $(\mathcal{X}, d_\infty)$ be the input space $\mathcal{X}$ with the infinity distance metric $d_\infty(x, x') = \|x - x'\|_\infty$, and $\mathcal{B}_\epsilon[x] = \{x' \in \mathcal{X} \mid d_\infty(x, x') \leq \epsilon\}$ be the closed ball of radius $\epsilon > 0$ centered at $x \in \mathcal{X}$. $\epsilon$ is also denoted as adversarial budget. Let $\mathcal{D} = \{(x_i, y_i)\}_{i=1}^n$ be a downstream dataset, where $x_i \in \mathbb{R}^d$, $d$ is the dimensionality of the input data, and $y_i \in \mathcal{Y} = \{0, 1, ..., C - 1\}$ is the ground-truth label. Let $f_{\theta_1} : \mathbb{R}^d \to \mathbb{R}^v$ be a pre-trained feature extractor parameterized by $\theta_1 \in \mathbb{R}^{d \times v}$ where $v$ is the dimensionality of the hidden features, $g_{\theta_2} : \mathbb{R}^v \to \mathbb{R}^z$ be a randomly-initialized linear classifier parameterized by $\theta_2 \in \mathbb{R}^{v \times z}$ where $z = |\mathcal{Y}|$ is the dimensionality of the predicted logits. For notational simplicity, we denote $h_\theta(\cdot) = g_{\theta_2} \circ f_{\theta_1}(\cdot)$ where $\theta = \{\theta_1, \theta_2\}$.

**Vanilla RFT (Zhang et al., 2019).** The training loss of vanilla RFT is formulated as follows:

$$\mathcal{L}_{\text{vanilla}}(\mathcal{D}; \theta, \beta) = \frac{1}{n} \sum_{(x,y) \in \mathcal{D}} \left\{ \underbrace{\ell_{\text{CE}}(h_\theta(x), y)}_{\textit{natural objective}} + \underbrace{\beta \cdot \ell_{\text{KL}}(h_\theta(\tilde{x}), h_\theta(x))}_{\textit{adversarial objective}} \right\}, \tag{1}$$

where $\beta > 0$ is a scalar of the adversarial objective, $\ell_{\text{CE}}(\cdot, \cdot)$ is the Cross-Entropy (CE) loss function, $\ell_{\text{KL}}(\cdot, \cdot)$ is the Kullback–Leibler (KL) loss function, and $\tilde{x}$ is the adversarial data generated via projected gradient descent (PGD) (Madry et al., 2018). Natural and adversarial objectives are used to learn the features of natural and adversarial data, respectively. In our paper, following Liu et al. (2023), the adversarial data $\tilde{x}$ is generated by maximizing the CE loss using PGD, i.e., $\tilde{x} = \arg \max_{\tilde{x} \in \mathcal{B}_\epsilon[x]} \ell_{\text{CE}}(h_\theta(\tilde{x}), y)$.

**TWINS (Liu et al., 2023).** TWINS proposed to combine vanilla RFT with a dual BN (Xie et al., 2020) framework to take advantage of pre-trained statistics, whose loss function is shown below:

$$\mathcal{L}_{\text{TWINS}}(\mathcal{D}; \theta, \beta, \gamma) = \mathcal{L}_{\text{vanilla}}(\mathcal{D}; \bar{\theta}, \beta) + \gamma \cdot \mathcal{L}_{\text{vanilla}}(\mathcal{D}; \theta, \beta), \tag{2}$$

where $\beta > 0$ and $\gamma > 0$ are hyperparameters. During conducting TWINS, all the parameters of $\theta$ are adaptively updated; the BN statistics of $\bar{\theta}$ are frozen as the pre-trained statistics and other parameters except for BN statistics of $\bar{\theta}$ are copied from $\theta$. Note that the adversarial data $\tilde{x}$ is generated according to the parameters $\theta$ via PGD.

## 3.2 Issues of Vanilla RFT and TWINS

We empirically discover vanilla RFT and TWINS have the issue of the optimization directions of minimizing both adversarial and natural objectives being significantly divergent. This issue can make the optimization unstable, thus impeding obtaining robustness and making RFT sensitive to hyperparameters. To validate the aforementioned issue, we calculate the gradient similarity (GS) as the cosine similarity between the gradient of the natural objective and that of the adversarial objective w.r.t. the FE parameters. To be specific, given a data point $(x, y) \in \mathcal{D}$, the GS of vanilla RFT and TWINS w.r.t. the FE (i.e., $\theta_1$) are calculated as follows:

$$\text{GS}_{\text{vanilla}}(x, y; \theta, \beta) = \text{sim}\big(\nabla_{\theta_1} \ell_{\text{CE}}(h_\theta(x), y), \nabla_{\theta_1} \ell_{\text{KL}}(h_\theta(\tilde{x}), h_\theta(x))\big); \tag{3}$$

$$\text{GS}_{\text{TWINS}}(x, y; \theta, \beta, \gamma) = \text{sim}\Big(\nabla_{\theta_1}\big(\ell_{\text{CE}}(h_{\bar{\theta}}(x), y) + \gamma \cdot \ell_{\text{CE}}(h_\theta(x), y)\big),$$

$$\nabla_{\theta_1}\big(\ell_{\text{KL}}(h_{\bar{\theta}}(\tilde{x}), h_{\bar{\theta}}(x)) + \gamma \cdot \ell_{\text{KL}}(h_\theta(\tilde{x}), h_\theta(x))\big)\Big), \tag{4}$$

where $\text{sim}(\cdot, \cdot)$ is the cosine similarity function. The smaller the GS is, the more divergent the gradient direction of optimizing natural and adversarial adversarial is. We report the average GS over all training data of vanilla RFT (blue lines) and TWINS (green lines) on DTD-57 (Cimpoi et al., 2014) in Figure 1a as well as extensive datasets (Cimpoi et al., 2014; Wah et al., 2011) in Figure 2a (in Appendix B.1).

Noticed from Figures 1a and 2a, we can observe the GS achieved by vanilla RFT and TWINS is quite low, which indicates that optimizing both natural and adversarial objectives via the FE can make the optimization direction orthogonal and even conflicting, thus leading to optimization oscillation. As shown in Figures 1b and 2b, compared to TWINS, vanilla RFT yields worse adversarial robustness in downstream tasks while achieving a lower GS. It indicates that the divergent optimization direction can lead to a lower robust test accuracy, which impedes obtaining adversarial robustness.

Besides, the issue of unstable optimization makes vanilla RFT and TWINS sensitive to hyperparameters. In Appendix B.2, we empirically validate that vanilla RFT and TWINS are sensitive to hyperparameters such as the learning rate. To achieve superior performance in downstream tasks, the authors of TWINS (Liu et al., 2023) conducted a grid search to find appropriate hyperparameters for each downstream task, which is extremely time-consuming and inconvenient for practical usage.

## 4 AUTOLORA: AUTOMATED RFT DISENTANGLED VIA A LOW-RANK BRANCH

To mitigate the aforementioned issue, we propose to disentangle RFT via a low-rank branch and then introduce heuristic strategies for automating scheduling hyperparameters.

## 4.1 Disentangling RFT via a Low-Rank Branch

To resolve the issue caused by optimizing both adversarial and natural objectives via the FE, we propose to leverage an auxiliary branch to disentangle the optimization procedure of natural and adversarial objectives. Inspired by PEFT (Hu et al., 2021; Chavan et al., 2023), we introduce a low-rank (LoRa) branch as an auxiliary parameter-efficient branch composed of two rank decomposition matrices $\boldsymbol{B} \in \mathbb{R}^{d \times r_{\mathrm{nat}}}$ and $\boldsymbol{A} \in \mathbb{R}^{r_{\mathrm{nat}} \times v}$ where $r_{\mathrm{nat}} \in \mathbb{N}$ is the rank, $d$ and $v$ are the dimensionality of the input data and hidden features, respectively. Therefore, $\boldsymbol{B}\boldsymbol{A} \in \mathbb{R}^{d \times v}$ has the same size as the parameters of the FE (i.e., $\theta_1 \in \mathbb{R}^{d \times v}$).

To disentangle the RFT, we propose to optimize the natural and adversarial objectives via the LoRa branch and the FE, respectively. Thus, we formulate the loss function of the RFT disentangled via the LoRa branch as follows:

$$\mathcal{L}_{\mathrm{LoRa}}(\mathcal{D};\theta, \boldsymbol{A}, \boldsymbol{B}, \lambda_1, \lambda_2) = \frac{1}{n} \sum_{(x,y)\in\mathcal{D}} \left\{ \underbrace{\lambda_1 \cdot \ell_{\mathrm{CE}}(h_{\{\bar{\theta}_1+\boldsymbol{B}\boldsymbol{A},\theta_2\}}(x), y)}_{natural\ objective} \right.$$

$$\left. + \underbrace{(1 - \lambda_1) \cdot \ell_{\mathrm{CE}}(h_\theta(\tilde{x}), y) + \lambda_2 \cdot \ell_{\mathrm{KL}}(h_\theta(\tilde{x}), h_{\{\bar{\theta}_1+\boldsymbol{B}\boldsymbol{A},\theta_2\}}(x))}_{adversarial\ objective} \right\}, \qquad (5)$$

where $\lambda_1 \geq 0$ and $\lambda_2 \geq 0$ are the scalars, $\theta = \{\theta_1, \theta_2\}$ denotes all the trainable parameters composed of the FE parameters $\theta_1$ and the classifier parameters $\theta_2$, and $\bar{\theta}_1$ denotes that the parameters $\theta_1$ do not require gradients.

According to Eq. 5, RFT disentangled via the LoRa branch can separate the optimization procedure of adversarial and natural objectives. Thus, disentangled RFT will update the FE parameters $\theta_1$ only by minimizing the adversarial objective that aims to learn the features of adversarial data; whereas, the gradient incurred by the natural objective will affect the LoRa branch instead of the FE. Therefore, the auxiliary LoRA branch solves the issue of divergent optimization directions, thus being able to improve adversarial robustness.

Besides, the FE also indirectly learns the knowledge of the natural objective via distilling knowledge from the LoRa branch, so that it prevents RFT from degrading the generalization. We can regard the LoRa branch as the teacher model that learns features of natural data and provides high-quality natural soft labels for the student model (i.e., the FE). Thus, the KL loss term in Eq. 5, which is used to penalize the KL divergence between the adversarial logits and natural soft labels generated by the LoRa branch, can be regarded as the knowledge distillation loss. In this way, the FE can implicitly learn the knowledge of natural objectives from the LoRa branch, thus maintaining the standard generalization.

Note that our proposed disentangled RFT via a LoRa branch is still low-cost thanks to the parameter-efficient LoRa branch. We empirically verify that the LoRa branch only introduces a quite small amount of extra trainable parameters that are less than 5% of the FE parameters (i.e., $|\boldsymbol{A}| + |\boldsymbol{B}| < 0.05 \cdot |\theta_1|$ when $r_{\mathrm{nat}} \leq 8$ validated in Table 4). Besides, the auxiliary LoRa branch does not incur extra inference latency since we drop this LoRa branch and only use the parameters $\theta$ for predicting test data during inference. Therefore, disentangled RFT can be an efficient and effective strategy to improve adversarial robustness while maintaining generalization in downstream tasks.

## 4.2 Automating Scheduling Hyperparameters

In this subsection, we introduce heuristic strategies of automating scheduling the hyperparameters including the learning rate and the scalars $\lambda_1$ and $\lambda_2$. We demonstrate the algorithm of our proposed automated RFT disentangled via a LoRa branch (AutoLoRa) in Algorithm 1.

**Automated scheduler of the learning rate (LR) $\eta$.** Lin et al. (2019) analogized the adversarial example generation process to the neural model training process. Therefore, recent studies (Wang & He, 2021; Yuan et al., 2022) have taken a similar strategy to control the step size in the adversarial example generation process inspired by the scheduler of LR in the neural model training process in order to improve the convergence of adversarial attack. Note that the step size and the LR are used to adjust the change rate of the adversarial example and the model parameters, respectively. Conversely, we conjecture that we can adopt the strategy of adjusting the step size for scheduling the LR as well. AutoAttack (Croce & Hein, 2020) proposed an automated scheduler of the step size

---

**Algorithm 1** Automated RFT disentangled via a LoRa branch (AutoLoRa)

---

1: **Input:** Training set $D$, pre-trained feature extractor $\theta_1$, maximum training epoch $E$
2: **Output:** Adversarially robust model $\theta$
3: Initialize classifier parameters $\theta_2$, $\theta = \{\theta_1, \theta_2\}$, and LoRa branch $\boldsymbol{A} = \mathcal{N}(\boldsymbol{0}, \boldsymbol{\sigma})$ and $\boldsymbol{B} = \boldsymbol{0}$
4: Initialize learning rate $\eta = 0.01$, epoch $e = 0$, batch size $\tau = 128$, training flag $FLAG = True$
5: **while** $FLAG$ **do**
6:     Update scalars $\lambda_1$ and $\lambda_2$ according to Eq. 6 and Eq. 7, respectively
7:     **for** batch $m = 1, \ldots, \lceil |D|/\tau \rceil$ **do**
8:         Sample a minibatch $S_m$ from $D$
9:         Calculate training loss $\mathcal{L}^* = \mathcal{L}_{\mathrm{LoRa}}(S_m; \theta, \boldsymbol{A}, \boldsymbol{B}, \lambda_1, \lambda_2)$
10:        Update parameters $\theta \leftarrow \theta - \eta \cdot \nabla_\theta \mathcal{L}^*$, $\boldsymbol{A} \leftarrow \boldsymbol{A} - \eta \cdot \nabla_{\boldsymbol{A}} \mathcal{L}^*$, $\boldsymbol{B} \leftarrow \boldsymbol{B} - \eta \cdot \nabla_{\boldsymbol{B}} \mathcal{L}^*$
11:     **end for**
12:     **if** Condition 1 **or** Condition 2 **then**
13:        $\eta \leftarrow \eta/2$
14:     **end if**
15:     **if** $\eta < 1e-5$ **or** $e \equiv E-1$ **then**
16:        $FLAG \leftarrow False$
17:     **else**
18:        $e \leftarrow e+1$
19:     **end if**
20: **end while**

---

guided by the classification loss, which has been validated as effective in improving the convergence of adversarial attacks. Therefore, we use a similar strategy to automatically adjust the LR.

Here, we introduce our proposed dynamic scheduler of the LR $\eta$ based on the robust validation accuracy during RFT inspired by AutoAttack (Croce & Hein, 2020). We start with the LR $\eta^{(0)} = 0.01$ at Epoch 0 and identify whether it is necessary to halve the current LR at checkpoint epoch $c_j \in \mathbb{N}^+$ where $j \in \{1, \ldots, M\}$ and $M$ is the number of checkpoint epochs. For example, when maximum training epoch $E = 60$ and $M = 15$, we check the LR every 4 epochs, i.e., $c_j \in \{0, 4, 8, \ldots, 56\}$. Given a validation set $\mathcal{D}_{\mathrm{val}} = \{(x_i, y_i)\}_{i=1}^{n_{\mathrm{val}}}$ of $n_{\mathrm{val}}$ data points and maximum training epoch $E \in \mathbb{N}$, we set the following two conditions:

1. $\sum_{e=c_{j-1}}^{c_j-1} \mathbb{1}[\mathrm{RA}(\mathcal{D}_{\mathrm{val}}; \theta^{(e+1)}) < \mathrm{RA}(\mathcal{D}_{\mathrm{val}}; \theta^{(e)})] \leq 0.75 \cdot (c_j - c_{j-1})$;

2. $\eta^{(c_{j-1})} \equiv \eta^{(c_j)}$ and $\mathrm{RA}(\mathcal{D}_{\mathrm{val}}; \theta^{(c_{j-1})})_{\max} \equiv \mathrm{RA}(\mathcal{D}_{\mathrm{val}}; \theta^{(c_j)})_{\max}$,

where $\mathbb{1}[\cdot]$ is an indicator function, $\theta^{(e)}$ refers to the parameters at Epoch $e \in \{0, 1, \ldots, E-1\}$, $\mathcal{D}_{\mathrm{val}}$ denotes a validation set, $\mathrm{RA}(\mathcal{D}_{\mathrm{val}}; \theta^{(e)})$ refers to the robust accuracy (RA) evaluated on the adversarial validation data using the parameter $\theta^{(e)}$, $\mathrm{RA}(\mathcal{D}_{\mathrm{val}}; \theta^{(c_j)})_{\max}$ denotes the highest robust validation accuracy found until $c_j$ epochs.

Condition 1 says that we halve the LR if the robust validation accuracy does not increase in the 75% stage between two adjacent checkpoint epochs (i.e., $c_{j-1}$ and $c_j$). Condition 2 says that we halve the LR if both the LR and the best robust validation accuracy remain the same at two adjacent checkpoint epochs. If at a checkpoint epoch $c_j$, the LR gets halved, then we take the parameters of the checkpoint that achieves the best robust validation accuracy (i.e., $\mathrm{RA}(\mathcal{D}_{\mathrm{val}}; \theta^{(c_j)})_{\max}$) as the initialization at next epoch.

**Automated scheduler of the scalars $\lambda_1$ and $\lambda_2$.** Given a training set $\mathcal{D}_{\mathrm{train}} = \{(x_i, y_i)\}_{i=1}^n$ of $n$ training data points, we set

$$\lambda_1^{(e)} = 1 - \mathrm{SA}(\mathcal{D}_{\mathrm{train}}; \{\theta_1^{(e)} + \boldsymbol{B}^{(e)} \boldsymbol{A}^{(e)}, \theta_2^{(e)}\})^\alpha, \tag{6}$$

$$\lambda_2^{(e)} = \lambda_2^{\max} \cdot \mathrm{SA}(\mathcal{D}_{\mathrm{train}}; \{\theta_1^{(e)} + \boldsymbol{B}^{(e)} \boldsymbol{A}^{(e)}, \theta_2^{(e)}\})^\alpha, \tag{7}$$

where $\lambda_1^{(e)}$ and $\lambda_2^{(e)}$ denote the weight terms at Epoch $e$, $\lambda_2^{\max} \geq 0$ is a hyper-parameter that controls the scale of $\lambda_2 \in [0, \lambda_2^{\max}]$, $\mathrm{SA}(\mathcal{D}_{\mathrm{train}}; \{\theta_1^{(e)} + \boldsymbol{B}^{(e)} \boldsymbol{A}^{(e)}, \theta_2^{(e)}\})$ refers to the standard accuracy (SA) of natural training data at Epoch $e$ evaluated via the LoRa branch, and $\alpha > 0$ is used for sharpening

Table 1: Performance benchmarks using ResNet-18. "SA" refers to the standard test accuracy. "PGD-10" and "AA" refer to the robust test accuracy evaluated by PGD-10 and AutoAttack, respectively. We report p-values of t-tests in Table 7 to justify the significance of performance gain.

| ResNet-18 | | Vanilla RFT | TWINS | AutoLoRa (ours) | Diff. (ours vs. Vanilla RFT) | Diff. (ours vs. TWINS) |
|---|---|---|---|---|---|---|
| CIFAR-10 | SA | 81.51 | 85.27 | *84.20* | **+2.69** | -1.07 |
| | PGD-10 | 52.97 | 53.04 | **54.27** | **+1.30** | **+1.23** |
| | AA | 47.52 | 47.89 | **48.95** | **+1.43** | **+1.06** |
| CIFAR-100 | SA | 58.55 | 63.03 | *62.10* | **+3.55** | -0.93 |
| | PGD-10 | 30.08 | 30.37 | **32.71** | **+2.63** | **+2.34** |
| | AA | 24.07 | 25.45 | **27.48** | **+3.41** | **+2.03** |
| DTD-57 | SA | 47.86 | 49.07 | *48.72* | **+0.86** | -0.35 |
| | PGD-10 | 23.49 | 24.87 | **27.34** | **+3.85** | **+2.47** |
| | AA | 20.37 | 20.87 | **21.91** | **+1.54** | **+1.04** |
| DOG-120 | SA | 47.23 | 48.73 | *48.57* | **+1.34** | -0.16 |
| | PGD-10 | 14.93 | 15.12 | **16.60** | **+1.67** | **+1.48** |
| | AA | 8.14 | 9.35 | **10.75** | **+2.61** | **+1.40** |
| CUB-200 | SA | 47.88 | 51.62 | *51.00* | **+3.12** | -0.62 |
| | PGD-10 | 21.13 | 21.35 | **22.70** | **+1.57** | **+1.35** |
| | AA | 15.66 | 15.95 | **16.62** | **+0.96** | **+0.67** |
| Caltech-256 | SA | 61.85 | 65.22 | *64.16* | **+2.31** | -1.06 |
| | PGD-10 | 43.22 | 43.36 | **44.13** | **+0.90** | **+0.76** |
| | AA | 38.17 | 38.23 | **39.00** | **+0.83** | **+0.77** |

the SA inspired by Zhu et al. (2021). As the training progresses with gradually increased standard training accuracy, $\lambda_1$ and $\lambda_2$ will decrease and increase, respectively.

Next, we provide the explanations for our design of the scheduler from the perspective of graduated optimization (Hazan et al., 2016; Hulse et al., 2019). The graduated optimization proposes to address a challenging optimization problem by first tackling a significantly simplified version, and gradually transforming the problem through optimization steps until it aligns with the complexity of the original optimization challenge. Our proposed scheduler gradually decreases $\lambda_1$ and increases $\lambda_2$, which lets the model first focus on optimizing the natural objective which is a simplified optimization problem, and then focus on solving the complex optimization problem of the adversarial objective in Eq. 5. In this way, our proposed scheduler improves the optimization of RFT, thus enhancing the performance.

## 5 EXPERIMENTS

In this section, we first conduct robustness benchmarks on various downstream tasks to validate the effectiveness of our proposed AutoLoRa shown in Algorithm 1. Then, we conduct ablation studies on various pre-trained backbones, the rank $r_{nat}$, the adversarial budgets $\epsilon_{pt}$ during robust pre-training, the sharpening hyperparameter $\alpha$, and the automated scheduler of LR.

**Baselines.** We take vanilla RFT (Zhang et al., 2019; Jiang et al., 2020) and TWINS (Liu et al., 2023) as the baseline methods. As for configurations of the learning rate and the scalars $\beta$ and $\gamma$, we exactly follow TWINS and also provide the detailed configurations in Table 5 (Appendix A).

**Pre-trained feature extractors.** In our work, we utilized ResNet-18 (He et al., 2016) and ResNet-50 that are adversarially pre-trained on ImageNet-1K (Deng et al., 2009) of $224 \times 224$ resolution. To be specific, we downloaded the pre-trained weights from the official GitHub of Salman et al. (2020). Following the settings of TWINS (Liu et al., 2023), we used pre-trained models with adversarial budget $\epsilon_{pt} = 4/255$ by default.

**Downstream tasks.** We considered six datasets as the downstream tasks. ❶ CIFAR-10 with 10 classes and ❷ CIFAR-100 (Krizhevsky, 2009) with 100 classes are low-resolution image datasets, whose training and test sets have 50,000 and 10,000 images, respectively. ❸ Describable textures dataset with 57 classes (DTD-57) (Cimpoi et al., 2014) is a collection of high-resolution textural images in the wild, which contains 2,760 training images and test 1,880 images. ❹ Stanford Dogs dataset with 120 dog categories (DOG-120) (Khosla et al., 2011) contains 12,000 training images and 8,580 test images. ❺ Caltech-UCSD Birds-200-2011 with 200 categories of birds (CUB-200) (Wah et al., 2011) is a high-resolution bird image dataset for fine-grained image classification, which contains 5,994 training images and 5,794 validation images. ❻ Caltech-256 with 257

Table 2: Performance benchmarks using ResNet-50. We report p-values of t-tests in Table 7 to justify the significance of performance gain.

| ResNet-50 | | Vanilla RFT | TWINS | AutoLoRa (ours) | Diff. (ours vs. Vanilla RFT) | Diff. (ours vs. TWINS) |
|---|---|---|---|---|---|---|
| CIFAR-10 | SA | 86.33 | 86.50 | *86.93* | **+0.60** | **+0.43** |
| | PGD-10 | 56.51 | 56.77 | **57.16** | **+0.65** | **+0.39** |
| | AA | 51.67 | 51.73 | **52.00** | **+0.33** | **+0.27** |
| CIFAR-100 | SA | 63.25 | 65.52 | *66.20* | **+2.95** | **+0.68** |
| | PGD-10 | 33.25 | 33.79 | **35.25** | **+2.00** | **+1.46** |
| | AA | 28.26 | 28.37 | **29.50** | **+1.24** | **+1.13** |
| DTD-57 | SA | 52.99 | 54.22 | *53.35* | **+0.36** | -0.87 |
| | PGD-10 | 27.36 | 28.94 | **30.21** | **+2.85** | **+1.28** |
| | AA | 23.35 | 23.88 | **25.59** | **+2.24** | **+1.71** |
| DOG-120 | SA | 62.68 | 63.64 | *62.33* | **+0.15** | -1.31 |
| | PGD-10 | 24.87 | 24.98 | **25.32** | **+0.45** | **+0.34** |
| | AA | 12.73 | 14.41 | **17.44** | **+4.71** | **+3.03** |
| CUB-200 | SA | 57.27 | 63.58 | *62.78* | **+5.51** | -0.81 |
| | PGD-10 | 28.37 | 29.60 | **31.07** | **+2.70** | **+1.47** |
| | AA | 23.09 | 23.71 | **24.40** | **+1.31** | **+0.69** |
| Caltech-256 | SA | 66.78 | 69.41 | *69.85* | **+3.07** | **+0.44** |
| | PGD-10 | 47.76 | 47.79 | **47.82** | **+0.06** | **+0.03** |
| | AA | 42.17 | 42.27 | **42.63** | **+0.46** | **+0.36** |

classes (Griffin et al., 2007) is a high-resolution dataset composed of 42,099 images in total. We randomly split it into 38,550 training data and 3,549 test data. Following Liu et al. (2023), we resized the images from both low-resolution image datasets (CIFAR-10 and CIFAR-100) and high-resolution datasets (DTD-57, DOG-120, CUB-200, Caltech-256) to $224 \times 224$ resolution. In this way, the input sizes are the same for pre-training and fine-tuning.

**Training configurations.** For the fair comparison, we set maximum training epoch $E = 60$ following (Liu et al., 2023). We used SGD as the optimizer, froze the weight decay of SGD as $1e - 4$, and set the rank of the LoRa branch $r_{nat} = 8$ by default. We set $\alpha = 1.0$ and $\lambda_2^{max} = 6.0$ by default. We randomly selected 5% of the entire training data as the validation set. During training, we used PGD-10 with an adversarial budget of $8/255$ and step size of $2/255$ to generate the adversarial training and validation data.

**Evaluation metrics.** We take standard test accuracy (SA) as the measurement of the generalization ability in downstream tasks. To evaluate the adversarial robustness, we use robust test accuracy evaluated by PGD-10 and AutoAttack (AA) (Croce & Hein, 2020) of the adversarial budget being $8/255$. For each method, we select the checkpoint that has the best PGD-10 test accuracy as the best checkpoint and report the performance of this best checkpoint in our results. We repeated the experiments 3 times and then conducted t-tests between the results of baselines (i.e., vanilla RFT and TWINS) and the results of our proposed AutoLoRa. We report the p-value of the t-test in Table 7 (Appendix B.3), which validates the significance of the improvement achieved by our AutoLoRa.

## 5.1 ROBUSTNESS BENCHMARKS ON VARIOUS DOWNSTREAM TASKS

In Tables 1 and 2, we demonstrate the performance benchmarks on six downstream tasks achieved by ResNet18 and ResNet-50, respectively. We annotate the robust test accuracy achieved by our proposed AutoLoRa in bold with underlining in Table 1 and 2. We can observe that AutoLoRa consistently obtains a higher robust test accuracy under both PGD-10 and AA for each downstream task and each backbone. Besides, the p-values obtained by t-tests in Table 7 (Appendix B.3) further validate that our improvement in adversarial robustness is significant. Notably, even compared with the previous SOTA method TWINS Liu et al. (2023), AutoLoRa achieves a 2.03% (from 25.45% to 27.48%) robustness gain using ResNet-18 on CIFAR-100 and a 3.03% (from 14.41% to 17.44%) robustness gain using ResNet-50 on the DOG-120 task.

## 5.2 ABLATION STUDY

In this subsection, we conducted ablation studies on the various pre-trained backbones, rank $r_{nat}$, the adversarial budgets $\epsilon_{pt}$ during robust pre-training, the sharpening hyperparameter $\alpha$, and the automated scheduler of LR.

Table 3: We report the performance benchmarks using vision transformers including ViT (Dosovitskiy et al., 2020) and DeiT (Touvron et al., 2021) on the CIFAR-10 dataset. SA and RA refer to standard test accuracy and PGD-10 robust test accuracy, respectively.

| RFT method | ViT (S/16) | | ViT (B/16) | | DeiT (DeiT-tiny) | | DeiT (DeiT-small) | |
|---|---|---|---|---|---|---|---|---|
| | SA | RA | SA | RA | SA | RA | SA | RA |
| Vanilla RFT | 80.31 | 51.06 | 83.87 | 53.39 | 78.85 | 49.72 | 81.73 | 51.92 |
| AutoLoRa | **80.97** | **51.51** | **84.79** | **54.10** | **79.49** | **50.52** | **82.33** | **52.63** |

Table 4: We report the effect of various ranks $r_{\text{nat}}$ on the performance of downstream tasks as well as the ratio of the LoRa branch's parameters to the original parameters (denoted as "Param. Ratio").

| Rank $r_{\text{nat}}$ | Param. Ratio | CIFAR-10 | | CIFAR-100 | | DTD-57 | | CUB-200 | |
|---|---|---|---|---|---|---|---|---|---|
| | | SA | RA | SA | RA | SA | RA | SA | RA |
| 2 | 1.30% | 84.89 | 52.71 | 61.86 | 32.00 | 45.85 | 25.43 | 52.04 | 22.06 |
| 4 | 2.49% | 83.69 | 52.75 | 62.91 | 32.75 | 45.81 | 24.95 | 50.50 | 22.51 |
| 8 | 4.87% | 84.20 | 54.27 | 62.10 | 32.71 | 48.72 | 27.34 | 51.00 | 22.70 |
| 16 | 9.62% | 84.87 | 54.17 | 61.81 | 32.63 | 49.15 | 27.23 | 50.43 | 21.92 |

**Various pre-trained backbones.** Table 3 demonstrates the performance of ImageNet (Deng et al., 2009) pre-trained vision transformers including ViT (Dosovitskiy et al., 2020) and DeiT (Touvron et al., 2021) achieved on the downstream CIFAR-10 dataset after RFT. The results validate that AutoLoRa is compatible with various pre-trained backbones and consistently achieves better generalization and adversarial robustness on downstream tasks compared to vanilla RFT.

**The rank $r_{\text{nat}}$** Table 4 shows that the test accuracy gradually rises in most cases as the rank $r_{\text{nat}}$ increases from 2 to 8, which indicates that a higher rank yields better robustness. The reason could be that a higher rank enables the LoRa branch to have more tunable parameters for fitting natural data and outputting higher-quality natural soft labels, which coincides with the discovery in Zhu et al. (2021) that high-quality soft labels are beneficial to improving performance. However, when $r_{\text{nat}} \geq 8$, the performance gain is marginal which means that $r_{\text{nat}} = 8$ is enough to capture the features of natural data and provide accurate natural soft labels. Therefore, we keep $r_{\text{nat}} = 8$ for the experiments in Section 5.1 by default.

**The adversarial budgets $\epsilon_{\text{pt}}$ during robust pre-training.** In Table 8, we report the performance using pre-trained FEs with different adversarial budgets $\epsilon_{\text{pt}} \in \{0, 1/255, 2/255, 4/255, 8/255\}$. The results show that larger $\epsilon_{\text{pt}}$ is beneficial to improving performance. Our proposed AutoLoRa achieves consistently better adversarial robustness than baselines.

**The automated scheduler of the LR.** We apply our proposed automated scheduler of LR into TWINS and report the performance in Table 9. We can observe that TWINS with an automated scheduler of LR can achieve a comparable performance compared to TWINS with tuned hyperparameters. It validates the effectiveness of our proposed automatic LR scheduler.

**The sharpening hyperparameter $\alpha$.** We report the performance under different $\alpha$ in Table 10. We can observe that both standard and robust test accuracy rise as $\alpha$ increases from 0.2 to 1.0 while the robust test accuracy begins to degrade as $\alpha$ increases from 1.0 to 5.0. It indicates that we do not need to sharpen the values of the standard test accuracy. Therefore, we keep $\alpha = 1.0$ by default.

## 6 CONCLUSIONS

This paper proposed an automated robust fine-tuning disentangled via a low-rank branch (AutoLoRa) that can automatically convert a pre-trained feature extractor to an adversarially robust model for the downstream task. We highlighted that vanilla RFT and TWINS have the issue where the gradient directions of optimizing both adversarial and standard objectives via the FE are divergent. This issue makes optimization unstable, thus impeding obtaining adversarial robustness and making RFT sensitive to hyperparameters. To solve the issue, we proposed a low-rank (LoRa) branch to make RFT optimize adversarial and standard objectives via the FE and the LoRA branch, respectively. Besides, we proposed heuristic strategies for automating the scheduling of the hyperparameters. Comprehensive empirical results validate that AutoLoRa can consistently yield state-of-the-art adversarial robustness in downstream tasks without carefully tuning hyperparameters. Therefore, AutoLoRa can be an automated and effective RFT framework which is significantly useful in practice. We leave how to conduct RFT to robustify LLMs against adversarial attacks (Xu et al., 2023a) and label-flipping attacks (Zhang et al., 2024) as the future work.

## ACKNOWLEDGEMENTS

This research is supported by the National Research Foundation, Singapore under its Strategic Capability Research Centres Funding Initiative. Any opinions, findings and conclusions or recommendations expressed in this material are those of the author(s) and do not reflect the views of National Research Foundation, Singapore.

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

## A  CONFIGURATIONS FOR BASELINES

We report the hyperparameters for reproducing the results of baselines. Note that we exactly followed the hyperparameters provided by Liu et al. (2023) since they are obtained by a time-consuming grid search. In our experiments, TWINS refer to the TRADES version of TWINS, known as TWINS-TRADES in Liu et al. (2023).

Table 5: The hyperparameter configurations of vanilla RFT and TWINS in our experiment following Liu et al. (2023). The format means $(\eta^{(0)}, \mathrm{WD}, \gamma)$ for TWINS and $(\eta^{(0)}, \mathrm{WD})$ for baselines where $\eta^{(0)}, \mathrm{WD}, \beta_2$ are the initial learning rate, the weight decay and the scalar. respectively.

| Method | CIFAR-10 | CIFAR-100 | DTD-57 | DOG-120 | CUB-200 | Caltech-256 |
|---|---|---|---|---|---|---|
| Vanilla RFT | (1e-2,1e-4) | (1e-3,1e-4) | (1e-2,1e-4) | (1e-3,1e-4) | (1e-2,1e-4) | (1e-2,1e-4) |
| TWINS | (1e-2,1e-4,1.0) | (1e-2,1e-4,1.0) | (1e-3,1e-4,1.0) | (3e-3,1e-4,1.0) | (1e-2,1e-4,3.0) | (3e-3,1e-4,1.0) |

## B  EXTENSIVE EMPIRICAL RESULTS

### B.1  EXTENSIVE RESULTS OF GRADIENT SIMILARITY AND ADVERSARIAL ROBUSTNESS

Here, we provide the extra results of gradient similarity and adversarial robustness evaluated on CIFAR-10 (Krizhevsky, 2009) and CUB-200 (Wah et al., 2011). The experimental details exactly follow Section 5.

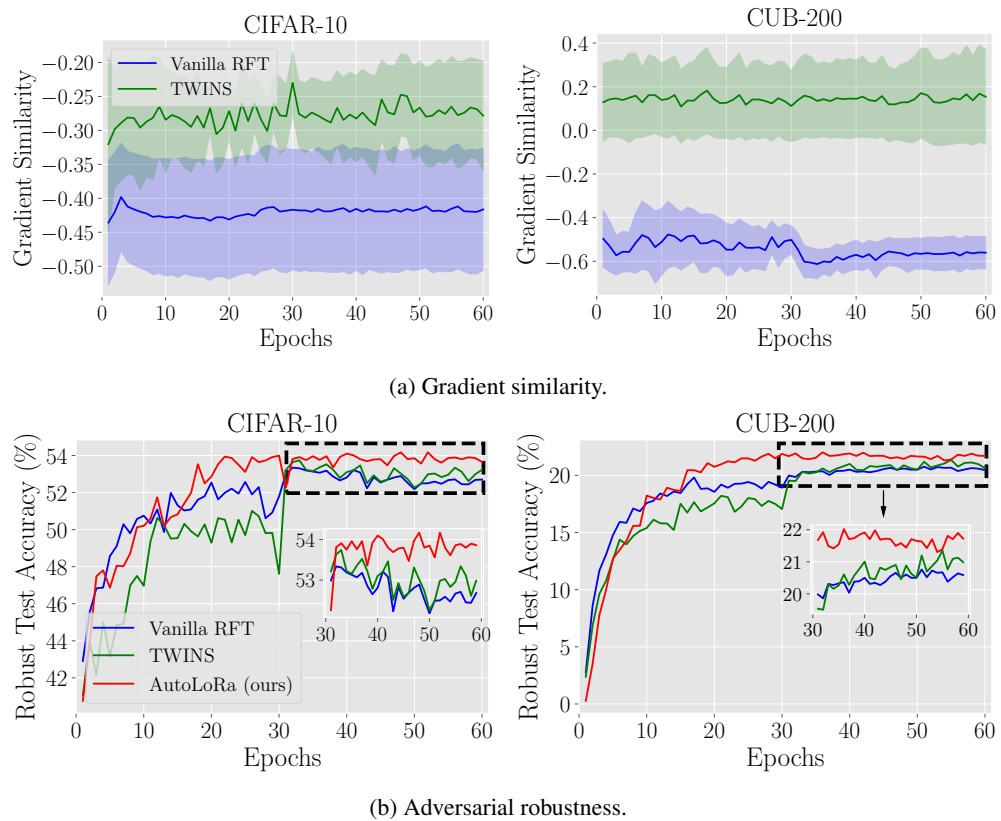

(a) Gradient similarity.

(b) Adversarial robustness.

Figure 2: Figure 2a shows the cosine similarity between the gradients of natural and adversarial objectives w.r.t. the feature extractor (FE). Figure 2b shows the robust test accuracy evaluated via PGD-10.

## B.2 SENSITIVITY TO HYPERPARAMETERS

We empirically show that vanilla RFT and TWINS are very sensitive to the hyperparameter (e.g., initial learning rate) caused by poor training stability in Table 6. We performed vanilla RFT and TWINS using the initial learning sampled from {0.001, 0.1, 0.03}. Table 6 shows that the robust accuracy (AA) ranges from 18.29 to 24.07 obtained by Vanilla RFT and 1.00 to 25.45 obtained by TWINS when the initial learning changes, which validates that vanilla RFT and TWINS are sensitive to hyperparameters.

Furthermore, we conducted AutoLoRa without the automated learning rate scheduler on CIFAR-100 using Resnet-18. The initial learning rate is sampled from {0.001, 0.1, 0.03} and is divided by 10 and 100 at Epoch 30 and 50, following Liu et al. (2023). Table 6 validates that the LoRa branch is effective in mitigating the sensitivity to the initial learning rate. It indicates that our proposed LoRa branch would be beneficial in decreasing the sensitivity to hyperparameters to some extent.

Table 6: Sensitivity to the initial learning rate.

| Initial learning rate | 0.001 | | 0.01 | | 0.03 | |
|---|---|---|---|---|---|---|
| | SA | RA | SA | RA | SA | RA |
| Vanilla RFT | 58.55 | 24.07 | 52.19 | 20.96 | 49.10 | 18.29 |
| TWINS | 61.54 | 23.51 | 63.03 | 25.45 | 1.00 | 1.00 |
| AutoLoRa without the automated learning rate scheduler | 61.97 | 27.51 | 62.88 | 27.21 | 62.21 | 27.33 |

## B.3 VALIDATING SIGNIFICANCE VIA T-TESTS

We repeated the experiments using the random seed from {0, 6, 66}. Therefore, for each downstream, each method has three results of SA, PGD-10, and AA, respectively. We conducted t-tests between the three results of SA/PGD-10/AA obtained by vanilla RFT and our proposed AutoLoRa as well as t-tests between the three results of SA/PGD-10/AA obtained by TWINS and our proposed AutoLoRa. We report the p-values obtained by t-tests in Table 7. Note that the p-value is smaller than 0.05, which means that the improvement gained by our proposed method is significant.

We annotate the p-value in bold when the p-value is smaller than 0.05 and the performance of AutoLoRa is better than the baseline. Table 7 validates that our proposed AutoLoRa achieves significant improvement in most cases.

Table 7: We report the p-values of t-tests between our proposed AutoLoRa (ours) and vanilla RFT as well as TWINS.

| Model | Method | P-value (vanilla RFT vs. ours) | | | P-value (TWINS vs. ours) | | |
|---|---|---|---|---|---|---|---|
| | | SA | PGD-10 | AA | SA | PGD-10 | AA |
| ResNet-18 | CIFAR-10 | **0.0001** | **0.0014** | **0.0020** | 0.0088 | **0.0002** | **0.0019** |
| | CIFAR-100 | **0.0010** | **8e-05** | **0.0008** | 0.0551 | **5e-05** | **0.0091** |
| | DTD-57 | 0.2393 | **0.0004** | **0.0037** | 0.4447 | **0.0036** | **0.0043** |
| | DOG-120 | 0.2568 | **0.0026** | **0.0002** | 0.4993 | **0.0043** | **0.0008** |
| | CUB-200 | **0.0215** | **0.0054** | **0.0051** | 0.8731 | **0.0244** | **0.0216** |
| | Caltech-256 | **0.0003** | **0.0023** | **0.0033** | 0.0458 | **0.0021** | **0.0026** |
| ResNet-50 | CIFAR-10 | **0.0019** | **0.0029** | **0.0056** | **0.0904** | **0.0341** | **0.0125** |
| | CIFAR-100 | **1e-05** | **3e-05** | **0.0024** | **0.0044** | **0.0004** | **0.0018** |
| | DTD-57 | **0.0320** | **9e-05** | **0.0002** | 0.0053 | **0.0031** | **0.0022** |
| | DOG-120 | 0.1595 | **0.0414** | **5e-06** | 0.0031 | **0.0329** | **6e-06** |
| | CUB-200 | **6e-06** | **5e-06** | **2e-06** | 0.1027 | **0.0010** | **0.0048** |
| | Caltech-256 | **1e-05** | **0.0457** | **0.0002** | **0.0151** | **0.0413** | **0.0050** |

## B.4 EMPIRICAL VALIDATION FOR AUTOLORA MITIGATING OPTIMIZATION DIVERGENCE

Given a data point $(x, y) \in \mathcal{D}$, the GS of AutoLoRa w.r.t. the FE (i.e., $\theta_1$) is calculated as follows:

$$\text{GS}_{\text{AutoLoRa}}(x, y; \{\theta, \boldsymbol{A}, \boldsymbol{B}\}) = \text{sim}\big(\nabla_{\theta_1}\ell_{\text{CE}}(h_\theta(\tilde{x}), y), \nabla_{\theta_1}\ell_{\text{KL}}(h_\theta(\tilde{x}), h_{\{\bar{\theta}_1 + \boldsymbol{BA}, \theta_2\}}(x))\big), \quad (8)$$

where $\text{sim}(\cdot, \cdot)$ is the cosine similarity function.

Figure 3 demonstrates the average GS over all training data of AutoLoRa on CIFAR-10, DTD-57, and CUB-200 datasets, respectively. We observe that, compared with the GS of Vanilla RFT and TWINS shown in Figures 1a and 2a, AutoLoRa significantly improves the GS. Thus, it validates that AutoLoRa alleviates the issue of divergent gradient directions.

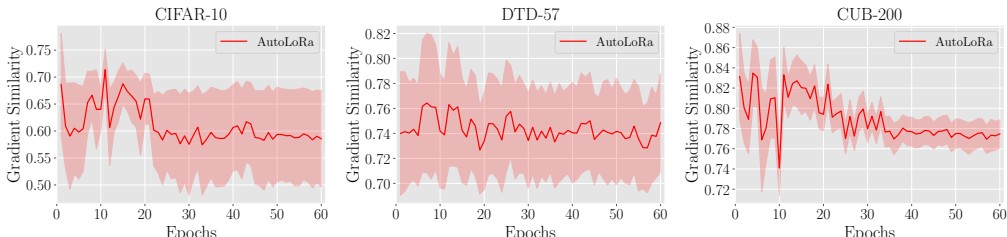

Figure 3: AutoLoRa mitigates the issue of divergent gradient directions.

## B.5 EXTENSIVE RESULTS OF ABLATION STUDIES

Table 8: The effect of adversarial budget $\epsilon_{\text{pt}}$ during robust pre-training. We keep adversarial budget as 8/255 during RFT and robustness evaluation.

| Datasets | | CIFAR-100 | | | | | DTD-57 | | | | |
|---|---|---|---|---|---|---|---|---|---|---|---|
| $\epsilon_{\text{pt}}$ | | 0 | 1/255 | 2/255 | 4/255 | 8/255 | 0 | 1/255 | 2/255 | 4/255 | 8/255 |
| SA | Vanilla RFT | 30.62 | 55.37 | 57.03 | 58.55 | 59.78 | 23.94 | **47.13** | 48.78 | 47.86 | 47.39 |
| | TWINS | **57.16** | **60.06** | **61.88** | **63.03** | 62.51 | 25.70 | 45.48 | **49.52** | **49.07** | 47.73 |
| | AutoLoRa | 56.36 | 59.37 | 61.13 | 62.10 | **62.82** | **26.54** | 46.22 | 48.56 | 48.72 | **47.76** |
| RA | Vanilla RFT | 13.33 | 26.33 | 28.28 | 30.08 | 31.44 | 7.23 | 23.62 | 25.32 | 23.49 | 25.48 |
| | TWINS | 26.24 | 27.67 | 28.33 | 30.37 | 31.83 | 8.72 | 20.32 | 24.15 | 24.87 | 25.16 |
| | AutoLoRa | **27.21** | **29.97** | **31.60** | **32.71** | **33.31** | **9.63** | **24.41** | **26.38** | **27.34** | **27.63** |

Table 9: The effect of the automated scheduler of LR on TWINS.

| Automatic LR scheduler | | w/o | w/ | diff. |
|---|---|---|---|---|
| CIAFR-10 | SA | 85.27 | 85.43 | +0.16 |
| | RA | 53.04 | 52.98 | -0.06 |
| CIFAR-100 | SA | 63.03 | 63.66 | +0.63 |
| | RA | 30.37 | 30.33 | -0.04 |
| DTD-57 | SA | 49.07 | 51.17 | +2.10 |
| | RA | 24.87 | 23.56 | -1.31 |
| Caltech-256 | SA | 65.22 | 64.53 | -0.69 |
| | RA | 43.36 | 42.98 | -0.38 |

Table 10: The effect of $\alpha$ on AutoLoRa.

| $\alpha$ | CIFAR-10 | | CIFAR-100 | |
|---|---|---|---|---|
| | SA | RA | SA | RA |
| 0.2 | 81.97 | 53.07 | 60.89 | 32.68 |
| 0.5 | 83.16 | 54.35 | 61.51 | 33.57 |
| 0.8 | 84.12 | 54.06 | 61.89 | 32.92 |
| 1.0 | 84.20 | 54.27 | 62.10 | 32.71 |
| 2.0 | 84.17 | 53.06 | 62.66 | 31.39 |
| 3.0 | 85.08 | 52.91 | 64.80 | 31.08 |
| 5.0 | 84.57 | 51.92 | 61.91 | 20.21 |

