# OpenReview forum: "AutoLoRa: An Automated Robust Fine-Tuning Framework"
_ICLR.cc/2024/Conference — ICLR 2024 poster_

### Official Review · Reviewer_qytY · 2023-10-28

**Soundness:** 2 fair
**Presentation:** 3 good
**Contribution:** 2 fair
**Rating:** 5
**Confidence:** 4

**Summary:**

**Summary Of The Paper:**

This paper introduces AutoLoRa, a parameter-free automated robust fine-tuning framework to improve adversarial robustness in downstream tasks that disentangles the optimization process into two distinct components: (1) optimizing natural objectives via the LoRa branch, (2) and adversarial objectives via the FE. It addresses the issue of divergent gradient directions, when optimizing both adversarial and natural objectives through the feature extractor (FE), in existing robust fine-tuning methods and achieves state-of-the-art results across various tasks without the need for hyperparameter tuning.

**Strengths:**

**Strength:**

-   The motivation that optimizing both adversarial and natural objectives through feature extractor yields divergent gradient directions make sense. The proposed disentangling of the training objective by introducing the LoRA branch is consistent with the motivation.
-   Extensive empirical evaluation (including the P-test) demonstrates the improvements in Robust Fine-Tuning on various tasks.

**Weaknesses:**

**Weakness:**

-   The reason for automating scheduling hyper-parameters is not well illustrated, and the ablation study in Table 5 can not show its superiority, especially for the *RA* metric.
-   In Formula 7, the constant factor $6$ is not well-explained, and it could be considered as a hyper-parameter with further ablation study.
-   Table 4 is confusing. Specifically, when the adversarial budget is set to $8$ which is the default configuration, the resulting metric is supposed to be that in Table 2. However, this is not true.
-   Typo in Section 5.2 Ablation Study, the end of adversarial budgets paragraph says "consistently achieves consistently".
-   Diverse network backbone architectures are encouraged to be considered beyond ResNet.

**Questions:**

Refer to the weakness section.

---

> ### Author Response · Authors · 2023-11-13
> **Response to Reviewer qytY**
>
> Many thanks for your comments! Please find our replies below.
>
> > [Reply to W1.1] TWINS requires expensive computation to grid-search the appropriate hyperparameters, which motivates us to automate the scheduler of hyper-parameters.
>
> TWINS [1] is sensitive to hyper-parameters, such as the initial learning rate (LR) as shown in the following table.
>
> |Initial LR for TWINS| SA | AA|
> |-|-|-|
> |0.001| 61.54 | 23.51 |
> |0.01| 63.03 | 25.45 |
> |0.03 | 1.00 | 1.00 |
>
> Therefore, the authors of TWINS [1] conducted a time-consuming grid-search for hyper-parameters including the initial LR $\eta^{(0)} \in \\{3e-4,1e-3,3e-3,1e-2,3e-2\\}$, the weight decay $\mathrm{WD} \in \\{1e-5,1e-4,1e-3,1e-2\\}$, and the scalar $\gamma \in \\{ 0.1,0.2,0.3,0.4,0.5,1.0\\}$, which aimed to achieve the state-of-the-art (SOTA) performance. It means that TWINS needs to repeat $120$ ($= 5 \times 4 \times 6$) runs of fine-tuning in order to achieve the SOTA performance for each backbone and each downstream task, which is extremely time-consuming and inconvenient for practical usage.
>
>
> By leveraging our proposed automatic scheduler, AutoLoRa can automatically achieve the SOTA performance by *ONE* run of robust fine-tuning on various models and various downstream tasks.
>
> > [Reply to W1.2] We make the first effort to propose an automated LR scheduler to avoid expensive grid-searching of the hyper-parameters.
>
> In Table 5, TWINS w/o LR scheduler carefully selects the appropriate hyper-parameters for each downstream task via expensive grid-search. Table 5 shows that TWINS w/ the LR scheduler can achieve comparable performance to TWINS w/o the LR scheduler, which validates the effectiveness of the LR scheduler in avoiding grid-search.
>
> > [Reply to W2] We provide extensive results under various factors in Eq. (7).
>
> We denote the factor as $\lambda_2^{\max} \geq 0$, i.e., $\lambda_2^{(e)} = \lambda_2^{\max} \cdot \mathrm{SA}(\mathcal{D}_{\mathrm{train}}; \\{\theta_1^{(e)}+B^{(e)}A^{(e)}, \theta_2^{(e)}\\})^\alpha$ (refer to Eq. (7)).
> There is a trade-off between standard accuracy and robust accuracy under AutoAttack (AA) by adjusting the factor $\lambda_2^{\max} $. We set $\lambda_2^{\max}=6.0$ by default to achieve good adversarial robustness without sacrificing too much standard generalization.
>
> |$\lambda_2^{\max}$ | SA | AA |
> |-|-|-|
> | 3.0 | 63.43 | 26.91 |
> | 6.0 | 62.10 | 27.48 |
> | 9.0 | 61.38 | 27.51 |
>
> The experiments are conducted on CIFAR-100 using ResNet-18.
>
> > [Reply to W3] In Table 4, we report performance using different adversarially pre-trained FEs under $\epsilon_{\mathrm{pt}} \in \\{ 0, 1/255,2/255, 4/255, 8/255\\}$. In Table 2, we report performance using the adversarially pre-trained FE under $\epsilon_{\mathrm{pt}} = 4/255$, which is the same setting as TWINS.
>
> $\epsilon_{\mathrm{pt}}$ refers to the adversarial budget used during the *pre-training* phase. We kept the adversarial budget as $8/255$ during the fine-tuning and evaluation phase.
> You can find that the results of $\epsilon_{\mathrm{pt}} = 4/255$ in Table 4 are consistent with the results in Table 2.
>
> > [Reply to W4] Thanks for pointing out this typo! We have revised it in our manuscript.
>
> > [Reply to W5] We provide extensive results on Wide ResNet of depth 28 and width 10 (WRN-28-10), which validate that AutoLoRa is compatible with various models.
>
> |WRN-28-10|SA|RA|
> |-|-|-|
> |Vanilla RFT| 61.56 | 35.64 |
> |AutoLoRa| **62.67** | **36.43**|
>
> We could not provide the results of TWINS because the authors of TWINS [1] did not implement TWINS on other models except for ResNet. We used the pre-trained WRN-28-10 on ImageNet-1K provided by [2] and applied Vanilla RFT and AutoLoRa on the CIFAR-100 task. *RA* refers to the robust test accuracy under PGD-20.
>
> ---
>
> > [Update **extensive results** for mitigating W5] Besides existing experiments on  *ResNet-18*, *ResNet-50*, and *WRN-28-10*, we provide results on extra pre-trained models, i.e., *Vision Transformer (ViT)* and *Data-Efficient Image Transformers (DeiT)* to validate the effectiveness of AutoLoRa on various model structures.
>
> 1. [ViT (S/16)](https://github.com/google-research/vision_transformer)
>
> |ViT|Standard accuracy|Robust accuracy|
> |-|-|-|
> |Vanilla RFT| 80.31 | 51.06 |
> |AutoLoRa| **80.97** | **51.51**|
>
> 2. ViT (B/16)
>
> |ViT|Standard accuracy|Robust accuracy|
> |-|-|-|
> |Vanilla RFT| 83.87 | 53.39 |
> |AutoLoRa| **84.79** | **54.10**|
>
> 3. [DeiT (DeiT-Tiny)](https://github.com/facebookresearch/deit)
>
> |DeiT|Standard accuracy|Robust accuracy|
> |-|-|-|
> |Vanilla RFT| 78.85 | 49.72 |
> |AutoLoRa| **79.49** | **50.52**|
>
> 4. DeiT (DeiT-small)
>
> |DeiT|Standard accuracy|Robust accuracy|
> |-|-|-|
> |Vanilla RFT| 81.73 | 51.92 |
> |AutoLoRa| **82.33** | **52.63**|
>
> ---
> *References:*
>
> [1] TWINS: A Fine-Tuning Framework for Improved Transferability of Adversarial Robustness and Generalization, CVPR 2023.
>
> [2] Using Pre-Training Can Improve Model Robustness and Uncertainty, ICML 2019.

---

> > ### Author Response · Authors · 2023-11-19
> > **We would like to know if you have any further questions or require additional clarification.**
> >
> > Dear **Reviewer qytY**,
> >
> > Thank you again for your valuable comments! We have carefully considered your comments and have provided our responses.
> >
> > Please let us know if our replies have satisfactorily addressed your concerns. Please do not hesitate to let us know if you have any further questions or if you require any additional clarification.
> >
> > Thank you very much!
> >
> > Best wishes,
> >
> > Authors

---

> > > ### Author Response · Authors · 2023-11-23
> > > **Your feedback is critical to us!**
> > >
> > > Dear **Reviewer qytY**,
> > >
> > > Thanks for your valuable comments! We have carefully considered your comments and provided our replies.
> > >
> > > Please let us know if our replies have satisfactorily addressed your concerns. Please do not hesitate to let us know if you have any further questions or if you require any additional clarification.
> > >
> > > Best wishes,
> > >
> > > Authors of Paper #538

---

### Official Review · Reviewer_4DPH · 2023-10-29

**Soundness:** 2 fair
**Presentation:** 3 good
**Contribution:** 3 good
**Rating:** 6
**Confidence:** 4

**Summary:**

Robust fine-tuning (RFT) is an efficient method to obtain a robust model on a downstream task by robustly fine-tuning a robust model that is adversarially trained on a large dataset.
However, this paper shows that the existing methods (vanilla RFT, TWINS [Liu et al. 2023]) suffer from the same issue that optimizing both adversarial and natural objectives yields significantly divergent gradient directions.
This paper points out that this divergence can hinder obtaining high robustness and make the training unstable.

To resolve the issue of the divergent gradient directions, this paper proposes a robust fine-tuning framework called AutoLoRa using the low-rank adaptation (LoRA) technique.
The idea is to have separate branches for adversarial and natural objectives: the adversarially pre-trained encoder is trained by the adversarial objective, and the LoRa branch is trained by the natural objective.

Additionally, AutoLoRa introduces automatic scheduling of hyperparameters, in contrast to vanilla RFT and TWINS, which require expensive hyperparameter searches. The balance between adversarial and natural objectives is determined by natural accuracy on the train set: as the standard accuracy increases, the weight on the natural objective decreases. The learning rate decays automatically with a condition of the validation accuracy.

In this paper, the pre-trained models are adversarially trained on ImageNet-1k.
On the six downstream datasets (CIFAR10, CIFAR100, DTD-57, DOG-120, CUB-200, and Caltech-256), AutoLoRa achieves higher adversarial robustness compared to vanilla RFT and TWINS.

**Strengths:**

S1. This paper points out the common issue of vanilla RFT and TWINS [Liu et al. 2023] that optimizing both adversarial and natural objectives yields significantly divergent gradient directions, which is beneficial knowledge for the adversarial robustness research community.

S2. The application of LoRA for the adversarial robustness problem is novel, and separating the branches for adversarial and natural objectives with the low-rank branch is interesting. The results show that their method can effectively improve adversarial robustness.

S3. The proposed automatic strategy to determine hyperparameters is useful since adversarial training is time-consuming.

**Weaknesses:**

W1. A more careful ablation study is needed.

- (W1-1.) It remains unclear how much each of the two proposed components contributes to the performance: (1) the automatic hyperparameter scheduling and (2) the LoRa branch. To clarify, does AutoLoRa without automatic scheduling (utilizing grid search) yield a similar result to AutoLoRa with dynamic hyperparameter scheduling? In other words, does the dynamic hyperparameter scheduling contribute to the robustness improvement, or is it only for avoiding grid search?

- (W1-2.) In line with (W1-1), it is also not evident how significantly the learning rate scheduling and the scalar parameter ($\lambda$) scheduling impact performance. While it appears that dynamic learning rate scheduling might not have the benefit of improving robustness, as seen in Table 5, I assume that dynamic scalar ($\lambda$) scheduling could contribute positively to performance, in addition to the benefit of avoiding grid search.

W2. The paper's claim regarding divergent gradient directions and training stability needs clarification.

- (W2-1.) The paper lacks evidence to support the claim that Vanilla RFT and TWINS are sensitive to hyperparameters. It would be helpful to specify which hyperparameters these methods are sensitive to and to what extent.
- (W2-2.) Since AutoLoRa employs automatic hyperparameter scheduling, it remains unverified whether the use of the LoRa branch indeed contributes to training stability regarding hyperparameters. To discuss the hyperparameter sensitivity, I would expect experiments comparing the different magnitudes of a specific hyperparameter and the corresponding performances for compared training methods.

W3: Natural accuracy trade-off in AutoLoRa compared to TWINS.
- For all cases in ResNet-18 and the three cases in ResNet-50, AutoLoRa exhibits a slightly lower natural accuracy compared to TWINS, despite achieving higher robust accuracy. Further discussion or insights on this trade-off would be valuable.

**Questions:**

Q1. Related to W1-2, it's worth considering the possibility of applying automated hyperparameter scheduling to Vanilla RFT and TWINS.   The scaler $\beta$ in Vanilla RFT or $\gamma$ in TWINS can be scheduled, by simply replacing $\lambda_2$ with $\beta$ or $\gamma$ in Equation 7. It would be interesting to see whether "Vanilla RFT + scaler scheduling" or "TWINS + scaler scheduling" can be better than the original methods. Additionally, comparing "Vanilla RFT + scaler scheduling" or "TWINS + scaler scheduling" with AutoLoRa could provide insights into the benefits of the LoRa branch.

Q2. How exactly is the gradient similarity calculated? A feature encoder has multiple layers to measure the gradient similarity.

Minor comment:
- It appears that TWINS in this paper corresponds to the TRADES version of TWINS, known as TWINS-TRADES [Liu et al. 2023]. It might help clarify the paper's context by explicitly mentioning this relationship.

-------
[Liu et al. 2023] Twins: A fine-tuning framework for improved transferability of adversarial robustness and generalization. CVPR2023

---

> ### Author Response · Authors · 2023-11-13
> **Response to Reviewer 4DPH (Part 1)**
>
> Many thanks for your comments! Please find our replies below.
>
> > [Reply to W1] We provide extensive results of AutoLoRa with or without the automated scheduler.
>
> |Automated LR scheduler|Automated scalar scheduler| SA | AA |
> |-|-|-|-|
> |&#10003;|&#10003;| 62.10 | 27.48 |
> |&#10003;|&#10005;| 61.96 | 27.21 |
> |&#10005;|&#10003;| 62.16 | 27.54 |
> |&#10005;|&#10005;| 61.64 | 27.29 |
>
> By comparing Row 1 with Row 2 in the above table, we can find that the automated scalar scheduler is beneficial to performance improvement and avoid the expensive grid-search.
>
> By comparing Row 1 with Row 3, we can find that the automated LR scheduler can help obtain the satisfied performance without expensive grid-searching for hyper-parameters.
>
> The above table uses ResNet-18 on the CIFAR-100 dataset. Without the automated scalar scheduler, we set $\lambda_1=0.5$ and $\lambda_2=6.0$. Without the automated LR scheduler, we set the initial learning rate as 0.01 and the learning rate is divided by 10 and 100 at Epoch 30 and 50.
>
> > [Reply to W2.1] We detailed the issue of sensitivity to hyper-parameters as follows.
>
> According to TWINS, Vanilla RFT and TWINS are senstive to the initial learning rate (LR), the weight decay, and the scalars. Therefore, the authors of TWINS conducted an expensive grid search on the aforementioned three hyper-parameters for Vanilla RFT and TWINS to obtain the best performance.
>
> The following results show that Vanilla RFT and TWINS are sensitive to the initial LR caused by poor training stability. For example, when the initial LR is sampled from \{0.001, 0.1, 0.03\}, the robust accuracy (AA) ranges from 18.29 to 24.07 obtained by Vanilla RFT and 1.00 to 25.45 obtained by TWINS.
>
> |Method|Initial LR| SA | AA|
> |-|-|-|-|
> |Vanilla RFT| 0.001| 58.55 | 24.07 |
> |Vanilla RFT| 0.01 | 52.19 | 20.96 |
> |Vanilla RFT| 0.03 | 49.10 | 18.29 |
> |TWINS| 0.001| 61.54 | 23.51 |
> |TWINS| 0.01| 63.03 | 25.45 |
> |TWINS| 0.03 | 1.00 | 1.00 |
> |AutoLoRa w/o automated LR scheduler| 0.001 | 61.97 | 27.51 |
> |AutoLoRa w/o automated LR scheduler| 0.01 | 62.88 | 27.21 |
> |AutoLoRa w/o automated LR scheduler| 0.03 | 62.21 | 27.33 |
>
> The experiments are conducted on CIFAR-100 using ResNet-18. The LR is divided by 10 and 100 at Epoch 30 and 50, following TWINS. Note that, TWINS fails to converge when the initial LR is 0.03.
>
> > [Reply to W2.2] We empirically validate that AutoLoRa mitigates the issue of sensitivity to hyper-parameters (e.g., the initial LR).
>
> We can find that AutoLoRa is less sensitive to hyper-parameters such as the initial LR since AutoLoRa's standard deviation of the accuracy among different initial LRs is very small compared to Vanilla RFT and TWINS.
>
> |Method|Standard deviation of SA|Standard deviation of AA|
> |-|-|-|
> |Vanilla RFT| 3.934 | 2.362 |
> |TWINS|28.896|11.097|
> |AutoLoRa|**0.385**|**0.123**|
>
> According to the table in **[Reply to W2.1]**, we show the standard deviation of the accuracy among different initial LRs in the above table.
>
>
> > [Reply to W3] The trade-off between robust and standard test accuracy could be mitigated by utilizing larger models. Further, we figure out a hyper-parameter (i.e., $\lambda_2^{\max}$ in Eq. (7)) that can adjust this trade-off.
>
> The trade-off between accuracy and adversarial robustness ubiquitously exists [1,2]. By comparing the results on CIFAR-10 and CIFAR-100 in Tables 1 and 2, we can find that utilizing larger models (that contain more trainable parameters) could mitigate the trade-off since both standard and robust accuracy get improved using ResNet-50.
>
> Further, we find that the hyper-parameter $\lambda_2^{\max}$ in Eq. (7), which is used for controlling the scale $\lambda_2^{(e)}$, can adjust the trade-off. We set $\lambda_2^{\max}=6.0$ by default in our paper. The following table shows that an appropriate $\lambda_2^{\max}$ for AutoLoRa (e.g., $\lambda_2^{\max}=3.0$) can yield improvement on both the standard and robust accuracy compared to TWINS.
>
> | | SA | AA |
> |-|-|-|
> |TWINS| 63.03 | 25.45 |
> | AutoLoRa ($\lambda_2^{\max}=3.0$) | **63.43** | 26.91 |
> | AutoLoRa ($\lambda_2^{\max}=6.0$) | 62.10 | 27.48 |
> | AutoLoRa ($\lambda_2^{\max}=9.0$) | 61.38 | **27.51** |
>
> The experiments are conducted on CIFAR-100 using ResNet-18.

---

> > ### Comment · Reviewer_4DPH · 2023-11-22
> > **Response**
> >
> > Regarding W1:
> > Thank you so much.
> > I observe that auto-scheduling mainly avoids the expensive grid search rather than improves accuracy.
> >
> > Regarding W2.1, W2.2:
> > The provided tables look good.
> > I recommend the authors add in the paper to clarify their claim that "Vanilla RFT and TWINS are hyperparameter-sensitive, and AutoLoRa is not."
> > However, I believe that only considering the initial learning rate is insufficient to claim "using LoRa branch reduces sensitivity to hyper-parameters."
> >
> > Regarding W3:
> > Thank you so much. I understand that AutoLoRa seems to require a capacity of network to mitigate clean-robust trade-off.

---

> > > ### Comment · Reviewer_4DPH · 2023-11-22
> > > **Response (part 2)**
> > >
> > > Regarding Q1:
> > > This table looks nice to validate the effectiveness of the proposed automated hyperparameter scheduling.
> > > Thank you so much.
> > >
> > > Regarding Q2.
> > > Thank you for the clarification. Layer-wise analysis could be interesting for future research.

---

> > > > ### Author Response · Authors · 2023-11-22
> > > > **Thanks for recognizing our responses!**
> > > >
> > > > Dear **Reviewer 4DPH**,
> > > >
> > > > Thanks for recognizing our responses! We are delighted to see our responses addressed your questions.
> > > >
> > > > Following your suggestions, we added the empirical results in Appendix A.5 of the updated manuscript to clearly show that vanilla RFT and TWINS have the issue of sensitivity to hyperparameters (e.g., the initial learning rate) and the effectiveness of AutoLoRa in mitigating this issue to some extent.
> > > >
> > > > Best wishes,
> > > >
> > > > Authors

---

> ### Author Response · Authors · 2023-11-13
> **Response to Reviewer 4DPH (Part 2)**
>
> > [Reply to Q1] We demonstrate the results of ''Vanilla RFT + scalar scheduling'' and "TWINS + scalar scheduling" as follows.
>
> The results validate that the scalar scheduler is applicable to Vanilla RFT and TWINS. Besides, it justifies the effectiveness of the auxiliary LoRa branch in enhancing performance.
>
>
> | RFT Method | SA | AA |
> |-|-|-|
> | Vanilla RFT | 58.55 | 24.07 |
> | Vanilla RFT + scalar scheduling | **59.09** | **24.43** |
> | TWINS | 63.03 | 25.45 |
> | TWINS + scalar scheduling | **63.14** | **25.87** |
> | AutoLoRa | 62.10 | 27.48 |
>
> The experiments are conducted on CIFAR-100 using ResNet-18. Following your suggestion, the scalar scheduler for $\lambda_2$ is applied to $\beta$.
>
> > [Reply to Q2] We provide a detailed illustration of calculating gradient similarity as follows.
>
> At each epoch, for each convolutional layer of the FE, we obtain a gradient similarity value according to Eq. (3) and (4). Then, we demonstrate the average and the standard deviation of all the convolutional layers' gradient similarity values in the FE at each epoch in Figure 1(a).
>
> > [Reply to Minor Comments] Thanks for your suggestion! We mention the relationship in Appendix A.1.
>
> *References:*
>
> [1] Theoretically Principled Trade-off between Robustness and Accuracy, ICML 2019.
>
> [2] Understanding and Mitigating the Tradeoff Between Robustness and Accuracy, ICML 2020.

---

### Official Review · Reviewer_kvxv · 2023-10-31

**Soundness:** 3 good
**Presentation:** 3 good
**Contribution:** 2 fair
**Rating:** 8
**Confidence:** 4

**Summary:**

The paper introduces AutoLoRa, an automated robust transfer learning framework. The authors present empirical evidence showing a significant divergence in the gradients of adversarial and natural objectives with respect to the feature extractor (FE), leading to unstable optimization. This observation motivates the authors to propose an auxiliary low-rank (LoRa) branch to disentangle the robust fine-tuning process, enabling the optimization of natural objectives through the LoRa branch and adversarial objectives through the FE.

Additionally, the authors introduce automatic schedulers for adjusting the learning rate and loss weights. The empirical results demonstrate that AutoLoRa achieves state-of-the-art robustness in downstream tasks without the need for hyperparameter search.

**Strengths:**

1. The paper is well-organized and well-written, making it easy to follow most parts of the paper.

2. The proposed method is well-motivated. The authors empirically discover that optimizing the natural and adversarial objectives leads to divergent optimization directions, which serves as the motivation for the LoRa branch.

3. The comprehensive results across various datasets provide strong support for the effectiveness of AutoLoRa.

4. AutoLoRa is parameter-free, offering practical utility. Additionally, the automated learning scheduler is adaptable to different methods.

**Weaknesses:**

1. The backbone models are pre-trained through robust supervised learning. It would be beneficial to demonstrate the performance of various backbone models pre-trained using robust self-supervised learning with AutoLoRa.

2. Robustness is currently assessed using AutoAttack. It would be more informative to assess the robustness under various attackers.

**Questions:**

Refer to Weaknesses.

---

> ### Author Response · Authors · 2023-11-13
> **Response to Reviewer kvxv**
>
> Many thanks for your comments! Please find our replies below.
>
> > [Reply to W1] AutoLoRa is applicable to fine-tuning the pre-trained models via self-supervisions.
>
> We applied AutoLoRa to finetune the pre-trained models via robust self-supervision including DynACL [1] and DynACL-AIR [2]. The pre-training and fine-tuning were conducted on CIFAR-10 and STL-10 datasets, respectively. We report the performance in the following table. Here, the robust accuracy is evaluated by AutoAttack.
>
> |Self-supervised method|Standard accuracy by Vanilla RFT|Robust accuracy by Vanilla RFT|Standard accuracy by AutoLoRa|Robust accuracy by AutoLoRa|
> |-|-|-|-|-|
> |DynACL-AIR [2]| 35.66 | 63.74 | **35.88** | **64.25** |
> |DynACL [1]| 35.25 | 63.53 | **35.51** | **64.16** |
>
> The results validate that AutoLoRa can improve both standard and robust test accuracy of the pre-trained models via self-supervisions in the downstream task.
>
> > [Reply to W2] AutoLoRa achieves better robustness under various attacks than Vanilla RFT and TWINS.
>
> We evaluated the robustness under three strong white-box attacks (APGD-CE [3], APGD-DLR [3] and FAB [4]) and one strong black-box attack (i.e., Square Attack [5]). We evaluate the robustness on CIFAR-10 of the pre-trained ResNet-18 after finetuning in the following table.
>
> |Method|APGD-CE [3]|APGD-DLR [3]|FAB [4]|Square Attack [5]|
> |-|-|-|-|-|
> |Vanilla RFT|51.91|49.05|47.94|55.81|
> |TWINS|51.76|49.46|48.40|56.08|
> |AutoLoRa|**52.06**|**49.91**|**48.87**|**56.38**|
>
> The experiments are conducted on CIFAR-10 using ResNet-18.
>
> *References:*
>
> [1] Rethinking the Effect of Data Augmentation in Adversarial Contrastive Learning, ICLR 2023.
>
> [2] Enhancing Adversarial Contrastive Learning via Adversarial Invariant Regularization, NeurIPS 2023.
>
> [3] Reliable evaluation of adversarial robustness with an ensemble of diverse parameter-free attacks, ICML 2020.
>
> [4] Minimally distorted adversarial examples with a fast adaptive boundary attack, ICML 2020.
>
> [5] Square attack: a query-efficient black-box adversarial attack via random search, ECCV 2020.

---

> > ### Comment · Reviewer_kvxv · 2023-11-21
> >
> > Hi Authors,
> >
> > Thanks for your responses and additional empirical results. After reading your responses, my concerns have been well
> > addressed.
> >
> > Besides, I have read other reviewers’ comments, and I am convinced that AutoLoRa is a powerful and automatic robust fine-tuning method, which can be a practical tool and a good addition to the community.
> > Furthermore, to my best of knowledge, the application of low-rank adaptation to robust fine-tuning is novel.
> >
> > Thus, I increase my score and confidence to support this paper.

---

> > > ### Author Response · Authors · 2023-11-22
> > > **Thanks for your support!**
> > >
> > > Dear **Reviewer kvxv**,
> > >
> > > We are delighted to hear that our responses have well addressed your questions. We appreciate your support by raising the rating.
> > >
> > > Best wishes,
> > >
> > > Authors

---

### Official Review · Reviewer_pJnK · 2023-10-31

**Soundness:** 3 good
**Presentation:** 3 good
**Contribution:** 2 fair
**Rating:** 6
**Confidence:** 3

**Summary:**

The paper proposes a decoupled fine-tuning framework for learning adversarially robust features. Specifically, a conventional robust finetuning pipeline consists of two losses: a natural objective and an adversarial objective. The paper shows that the divergence in gradients from the two objectives is correlated with downstream accuracy on robustness benchmarks. Therefore, it proposes to decouple a model into two branches where the second branch is constructed using Low-rank adaptation (LORA). In the decoupled training scheme, the main model (first branch) is only exposed to the adversarial objective and the LORA branch (second branch) is only exposed to the natural objective. The paper claims that the disentanglement avoids gradient divergence and leads to better downstream robustness.

**Strengths:**

* **Clear presentation**: the paper uses clear and concise notations for equations. The method section is easy to follow.

* **Good ablation study**: the paper conducts ablation study on important hyper-parameters such as the rank in LORA and the LR scheduler.

**Weaknesses:**

* **Missing ablation on Equation 5**: Equation 5 is the main loss function of the proposed method, which has three terms. The second cross-entropy term is a new addition $L_{KL}(h_{\theta}(\tilde{x}),y)$ in this paper and is not motivated and ablated in the experiments.
* **Doubt on mitigating divergence**: a main motivation of the method is that it can avoid divergent gradient updates on the main model parameters. However, this is not validated through experiments explicitly. For example, even though the main model parameters are not directly trained on the natural objective, it is indirectly affected by the natural objective through the KL divergence. Moreover, the unexplained second cross-entropy term $L_{KL}(h_{\theta}(\tilde{x}),y)$ can be seen as a natural objective on perturbated input.
* **Why parameter-free**: the claim on parameter-free can be confusing. The model not only fine-tunes the main model parameters but also additional LORA parameters. So, it is not parameter-free in the sense of fine-tuning. Even though the method reduces the need for extensive hyper-parameter tuning, the design choices of the automated scheduler and the rank selection for LORA are all hyper-parameters. It’s not clear what aspect of the proposed method is parameter-free.

**Questions:**

* Could the authors provide the cosine similarity between the two terms in the adversarial objective in Equation 5 and comment on the functionality of the second cross-entropy term $L_{KL}(h_{\theta}(\tilde{x}),y)$?

* Could the authors clarify the parameter-free characteristic? It could be that I misunderstood the meaning here.

---

> ### Author Response · Authors · 2023-11-13
> **Response to Reviewer pJnK**
>
> Many thanks for your comments! Please find our replies below.
>
> > [Reply to W1&Q1.2] The second cross-entropy (CE) term $\ell_{\mathrm{CE}}(h_{\theta}(\tilde{x},y))$ aims to provide *ground-truth labels* for the FE to learn the adversarial data, which can improve the adversarial robustness as validated in the following table.
>
> Without the second CE term, the FE would be only regulated by the KL term that uses the natural soft labels generated by the LoRa branch. However, the natural soft labels could contain wrongly classified labels, which could degrade the performance. Empirically, we found that the natural training error evaluated on the LoRa branch keeps about $30\\%$ on CIFAR-100, which indicates that there are some noisy labels in natural soft labels generated by the LoRa branch. Thus, it necessitates the second CE term to provide the ground-truth labels for updating the FE together, which can further improve performance.
>
> The following empirical results validate that *with* the second CE term, both adversarial robustness and standard generalization achieved by AutoLoRa get improved. The experiments are conducted on CIFAR-100 using ResNet-18.
>
> | | SA | AA |
> |-|-|-|
> | Without the second CE term (setting $\lambda_1 = 1.0$) | 61.84 | 27.07 |
> | With the second CE term (setting $\lambda_1 = 0.5$) | 61.96 | 27.21 |
> | With the second CE term (dynamic scheduling $\lambda_1$) | **62.10** | **27.48** |
>
> > [Reply to W2&Q1.1] AutoLoRa mitigates the issue of divergent gradient directions, which is further empirically validated in Figure 3 (in Appendix A.4).
>
> Figure 3 shows that AutoLoRa achieves high cosine similarity between the two terms of the adversarial objective in Eq. (5), which empirically justifies that AutoLoRa mitigates divergence.
>
> Note that both the second CE term and the KL term aim to let the FE learn the adversarial data. In this way, the LoRa branch disentangles the model to learn the adversarial and natural data via the FE and the LoRa branch, respectively. Thus, AutoLoRa can mitigate the issue of divergent gradient directions.
>
> > [Reply to Q2] The term *parameter-free* refers to that you do not need to specify any hyper-parameters (e.g., the initial learning rate, the learning rate scheduler, the scalars) when applying AutoLoRa to various downstream tasks (e.g., CIFAR, DTD-57, CUB-200) and various models (e.g., ResNet-18, ResNet-50).
>
> The term *parameter-free* is inspired by the title of AutoAttack [1], in which *parameter-free* refers to that you do not need to specify any hyper-parameters while utilizing the AutoAttack to attack various models on various datasets.
>
> AutoLoRa is *parameter-free* benefiting from our proposed automated scheduler for the learning rate and the scalars. We empirically validate that AutoLoRa is *parameter-free* since AutoLoRa only needs to specify the downstream dataset and the model, without specifying any other hyper-parameters, to automatically obtain superior performance in downstream tasks.
>
> *References:*
>
> [1] Reliable evaluation of adversarial robustness with an ensemble of diverse parameter-free attacks, ICML 2020.

---

> > ### Comment · Reviewer_pJnK · 2023-11-14
> > **Discussion**
> >
> > Thank you for the clarification. I appreciate the additional visualization on the cosine similarity between gradients of the AutoLORA method.
> >
> > Now I understand that the parameter-free property refers to the automatic hyper-parameter scheduling. However, I am still not completely convinced by the parameter-free claim in the paper.
> >
> > The hyper-parameter scheduler is an empirical function of the robust accuracy, standard accuracy, a hyper-parameter alpha (Eq.6,7), and another hyper-parameter that balances between the standard accuracy and the robust accuracy (Eq.7).
> >
> > First, the empirical nature of this scheduler makes it less convincing unless more experiments are provided, e.g., experiments beyond ResNet as suggested by Reviewer qytY. Second, even though the hyper-parameters in the scheduler use default values in existing experiments, there is no guarantee that they will perform well on other datasets and architectures. In other words, the validity of the parameter-free claim depends on what experiments are demonstrated in the paper and its generalizability cannot be tested.
> >
> > In my opinion, the scheduler is a good idea inspired by heuristics. It could be that with the proposed scheduler, the optimization becomes less sensitive to the choice of hyper-parameters and is easier to tune. While this weakens the claim in the paper, it is much more reasonable and still makes a good contribution.

---

> ### Author Response · Authors · 2023-11-16
> **We provide extensive results to strengthen our parameter-free claim.**
>
> First, many thanks for your commending that our heuristic scheduler is good and makes a good contribution!
>
> Second, following your suggestions, we provide results on extensive model structures and extensive downstream datasets to strengthen our parameter-free claim.
>
> > Besides existing experiments on  *ResNet-18* and *ResNet-50*, we provide results on three extra pre-trained models including *Wide ResNet (WRN), Vision Transformer (ViT)*, and *Data-Efficient Image Transformers (DeiT)* to strengthen our parameter-free claim.
>
> 1. [WRN with depth 28 and width 10](https://github.com/hendrycks/pre-training/tree/master/robustness/adversarial)
>
> |WRN|Standard accuracy|Robust accuracy|
> |-|-|-|
> |Vanilla RFT| 61.56 | 35.64 |
> |AutoLoRa| **62.67** | **36.43**|
>
> 2. [ViT (S/16)](https://github.com/google-research/vision_transformer)
>
> |ViT|Standard accuracy|Robust accuracy|
> |-|-|-|
> |Vanilla RFT| 80.31 | 51.06 |
> |AutoLoRa| **80.97** | **51.51**|
>
> 3. [DeiT (DeiT-tiny)](https://github.com/facebookresearch/deit)
>
> |DeiT|Standard accuracy|Robust accuracy|
> |-|-|-|
> |Vanilla RFT| 78.85 | 49.72 |
> |AutoLoRa| **79.49** | **50.52**|
>
> 4. [DeiT (DeiT-small)](https://github.com/facebookresearch/deit)
>
> |DeiT|Standard accuracy|Robust accuracy|
> |-|-|-|
> |Vanilla RFT| 81.73 | 51.92 |
> |AutoLoRa| **82.33** | **52.63**|
>
> ---
>
> **[Update extensive results]**
>
> 5.  [ViT (B/16)](https://github.com/google-research/vision_transformer)
>
> |ViT (B/16)|Standard accuracy|Robust accuracy|
> |-|-|-|
> |Vanilla RFT| 83.87 | 53.39 |
> |AutoLoRa| **84.79** | **54.10**|
>
> ---
>
> As for WRN, pre-training and fine-tuning are conducted on ImageNet-1K and CIFAR-100, respecitve. As for ViT and DeiT,pre-training and fine-tuning are conducted on ImageNet and CIFAR-10, respecitve. We do not provide the results of TWINS since TWINS is only implemented on ResNet.
>
> > Besides existing experiments on *CIFAR-10, CIFAR-100, DTD-57, DOG-120, CUB-200*, and *Caltech-256*, we provide results on three extra downstream dataests including *STL-10, Caltech-101*, and *Cars-196* to strengthen our parameter-free claim.
>
> 1. [STL-10](https://cs.stanford.edu/~acoates/stl10/)
>
> |STL-10| Standard accuracy|Robust accuracy|
> |-|-|-|
> |Vanilla RFT| 63.74 | 35.66 |
> |AutoLoRa| **64.25** |**35.88** |
>
> 2. [Caltech-101](https://data.caltech.edu/records/mzrjq-6wc02)
>
> |Caltech-101| Standard accuracy|Robust accuracy|
> |-|-|-|
> |Vanilla RFT| 84.72| 60.46 |
> |TWINS| 87.83 | 61.59 |
> |AutoLoRa| **90.01** |**62.29** |
>
> 3. [Cars-196](https://www.tensorflow.org/datasets/catalog/cars196)
>
> |Cars-196| Standard accuracy|Robust accuracy|
> |-|-|-|
> |Vanilla RFT | 65.89 | 37.21 |
> |TWINS| 68.48 | 39.69 |
> |AutoLoRa| **70.23** |**41.19** |
>
> As for STL-10, we used pre-trained ResNet-18 via adversarial contrastive learning on CIFAR-10 of $32 \times 32$ resolution and then fine-tuned the model on STL-10 of $32 \times 32$ resolution. We do not provide the results of TWINS since TWINS is only implemented on datasets of $224 \times 224$ resolution. As for Caltech-101 and Cars-196, we used ImageNet-1k pre-trained ResNet-50 for fine-tuning.

---

> > ### Author Response · Authors · 2023-11-19
> > **We would like to know if you have any further questions or require additional clarification.**
> >
> > Dear **Reviewer pJnK**,
> >
> > Thank you again for your valuable comments! We have carefully considered your comments and have provided our responses.
> >
> > Please let us know if our replies have satisfactorily addressed your concerns. Please do not hesitate to let us know if you have any further questions or if you require any additional clarification.
> >
> > Thank you very much!
> >
> > Best wishes,
> >
> > Authors

---

> > > ### Comment · Reviewer_pJnK · 2023-11-20
> > > **Post rebuttal remark**
> > >
> > > Hi Authors,
> > >
> > > I appreciate the additional empirical evidence and clarifications and have increased my score to reflect this.  However, I still feel that the claim on parameter-free is too strong given the empirical nature even though the rest of the paper is very good. As I said earlier, we can't be sure if the proposed heuristic, which has hyper-parameters of its own, will work well unless we test it. I wouldn't be surprised that if some tuning is needed to achieve the best performance in a slightly different setup. In other words, the proposed scheduler is not functionally parameter-free. It just means that the default hyper-parameters work well across the experiments in this paper, i.e., robust to hyper-parameters.
> > >
> > > In fact, I would recommend softening this claim if accepted because this would only make the paper more rigorous.

---

> > > > ### Author Response · Authors · 2023-11-21
> > > > **Thanks again for the follow-up comments.**
> > > >
> > > > Dear **Reviewer  pJnK**,
> > > >
> > > > First, we would like to express our sincere appreciation to you for recognizing our responses and increasing your score.
> > > >
> > > > Second, following your suggestions, we removed the claim word 'parameter-free' in our paper and updated our manuscript.
> > > >
> > > > Previously, the word 'parameter-free' only occurred in the Title and Conclusions. In detail, we made the following revisions:
> > > >
> > > > 1. Title: AutoLoRa: A Parameter-Free Automated Robust Fine-Tuning Framework -> AutoLoRa: An Automated Robust Fine-Tuning Framework
> > > >
> > > > 2. Conclusions: Therefore, our proposed AutoLoRa can be an effective and parameter-free RFT framework. -> Therefore, AutoLoRa can be an automated and effective RFT framework which is significantly useful in practice.
> > > >
> > > > Thank you again for your valuable comments!
> > > >
> > > > Best wishes,
> > > >
> > > > Authors

---

### Author Response · Authors · 2023-11-18
**We would like to know if you have any further questions or require additional clarification.**

Dear Reviewers,

Thank you for taking the time to review our work and for providing us with valuable feedback. We have carefully considered your comments and have provided our responses.

If you have any further questions or require additional clarification, please kindly let us know.

In particular, we would like to ask **Reviewer pJnK** and **Reviewer qytY** if our responses satisfactorily address their concerns.

Thank you again for your valuable input.

Best wishes,

Authors

---

### Meta-Review · Area_Chair_tDfU · 2023-12-05

**Metareview:**

This submission introduces a framework for Robust Fine-Tuning motivated by the empirical observation of divergent gradient directions for adversarial and natural objectives. This issue is addressed by introducing an architectural change in the form of a newly introduced LoRa branch and a subsequent allocation of trainable parameters to these objectives. An attractive feature of the method is also its efficiency improvement by an automated LR scheduler. In my own view, I would've liked to see more justification for LoRa as a specific choice, given the plethora of other Fine-Tuning methods available, given that the author considered this choice essential to include in the paper title.

The authors have done their utmost to respond to each reviewer, both providing concrete evidence for specific choices but also engaging and, on occasion, accepting relevant feedback (e.g. discussion around parameter-free formulation, which I agree with). As a result, I believe that the revised manuscript has improved and is deserving of acceptance at ICLR.

**Justification For Why Not Higher Score:**

Overall, I feel that a borderline score by 3/4 of reviewers would not have warranted a spotlight or oral recommendation. Personally, given its relative importance to this paper (literally in the title), I would've also liked to see more justification for LoRa as a parameter-efficient Fine-tuning method, given the plethora of other options.

**Justification For Why Not Lower Score:**

Among all reviewers, qytY is the only reviewer suggesting a rating slightly below the acceptance threshold. Reviewer qytY has regretfully not responded to the author's comments, which have at least partially addressed some of the criticism. As a result, I somewhat down-weighted the contribution of this reviewer's score in making my decision. As there is at least one reviewer with a high score, I'm happy for this to push the paper over the acceptance threshold, as this suggests a chance for positive feedback, at least from a subset of the ML community.

---

### Decision · Program_Chairs · 2024-01-16

Accept (poster)